# High-Risk Human Papillomavirus Detection via Cobas^®^ 4800 and REBA HPV-ID^®^ Assays

**DOI:** 10.3390/v14122713

**Published:** 2022-12-03

**Authors:** Sasiprapa Liewchalermwong, Shina Oranratanaphan, Wichai Termrungruanglert, Surang Triratanachat, Patou Tantbirojn, Nakarin Kitkumthorn, Parvapan Bhattarakosol, Arkom Chaiwongkot

**Affiliations:** 1Department of Microbiology, Faculty of Medicine, Chulalongkorn University, Bangkok 10330, Thailand; 2Medical Microbiology Interdisciplinary Program, Graduate School, Chulalongkorn University, Bangkok 10330, Thailand; 3Department of Obstetrics and Gynecology, Faculty of Medicine, Chulalongkorn University, Bangkok 10330, Thailand; 4Department of Oral Biology, Faculty of Dentistry, Mahidol University, Bangkok 10400, Thailand; 5Center of Excellence in Applied Medical Virology, Faculty of Medicine, Chulalongkorn University, Bangkok 10330, Thailand

**Keywords:** HPV testing, cervical cancer

## Abstract

Persistent infection with high-risk human papillomaviruses (HR-HPVs), particularly HPV16 and 18, has long been known to induce cervical cancer progression. However, given that a minority of HPV-infected women develop cancer, analysis of HR-HPV-infected women could help to predict who is at risk of acquiring cervical cancer. Therefore, to improve HR-HPVs detection, we used the FDA-approved cobas^®^ 4800 HPV and REBA HPV-ID^®^ HPV assays to detect HR-HPVs in colposcopy-derived cervical cells from 303 patients, detecting 72.28% (219) and 71.62% (217) of HR-HPVs positive cases, with HPV16 detection rates of 35.64% (108) and 30.69% (93), respectively. Of the HPV16-positive cases, cobas^®^ 4800 and REBA HPV-ID^®^ identified 28.81% (51) and 25.42% (45) of the CIN1 cases, and 55% (33) and 50% (30) of the 60 CIN2/3 cases, respectively. HPV-diagnostic concordance was 82.17% overall (kappa = 0.488), 87.45% for HR-HPVs (kappa = 0.689), and 88.33% for CIN2/3 (kappa = 0.51). The HR-HPVs detection rates of these assays were comparable. Our findings reveal that the FDA-approved HR-HPVs detection assay is appropriate for screening women with HR-HPVs infection, and for predicting increased risk of cervical cancer progression. REBA HPV-ID^®^ can be used to detect low risk-HPV types in high-grade cervical lesions that are HR-HPV negative as well as in the distribution of HPV types.

## 1. Introduction

Cervical cancer is the fourth most common cancer among women worldwide, with more than half a million estimated new cases and 311,365 deaths in 2018 (GLOBOCAN) [1]. High-risk human papillomaviruses (HR-HPVs) have been detected in 99.7% of cervical cancer cases worldwide [2], with HPV16 being the most prevalent (>50% of cases), and HPV18 occurring in about 20% of all cases. HPV16 and HPV18 have been strongly linked to high-grade cervical lesions [3]. Nonetheless, only a small proportion of HPV-infected women get cancer, and malignant transformation takes several years [3,4]. HR-HPV31, 33, 35, 45, 52 and 58 have also been detected in cervical cancers worldwide [2,5,6]. 

The conventional Pap smear has been used for cervical cancer screening for more than 50 years [7], with a false-negative rate of 20–30% [8]. Compared to the conventional Pap smear, liquid-based cytology provides a better sample quality, improving the low-grade squamous intraepithelial lesion (LSIL) diagnostic rate and abnormal-cell detection accuracy [9,10,11]. 

Given the high prevalence of HR-HPVs in cervical cancer, HR-HPV-DNA testing has been introduced as an additional test, in combination with cytology analysis, to screen women who need further colposcopy examination. HR-HPV testing reportedly provided more sensitive CIN2+ diagnosis than cytology [12,13]. HPV testing improved CIN1-3-detection in HR-HPV-positive women, with atypical squamous cells of undetermined significance (ASCUS) referred for colposcopy [14]. HPV16- and 18-positive samples with negative cytology were identified as CIN2 or more severe lesions via histology [15]. Conization of CIN2+ lesions in women with persistent HPV16 infection revealed a higher rate of recurrent cervical lesions [16,17]. HR-HPV testing is therefore essential in cervical cancer screening.

Although various HPV-DNA detection tests exist, the five US Food and Drug Administration (US FDA)-approved assays detect only the HR-HPV types [18,19]. It has been reported that HR-HPVs testing showed a significant sensitivity for detecting CIN2+ lesions [20,21,22]. Some national cervical cancer screening programs rely solely on HPV primary testing [23,24]. HR-HPV assays, which detect HPV16 and 18 in clinical samples, are commonly used in cervical cancer screening to triage women at higher risk of developing cancer, and who should be referred for colposcopy.

The US FDA-approved HPV-DNA detection assays include cobas^®^ 4800, Hybrid Capture 2 (HC2) and three others [18,19]. Most studies have compared cobas^®^ 4800 to the HC2 assay, revealing good concordance [21,25,26,27]. One study reported that the cobas^®^ 4800 assay showed higher sensitivity, specificity, and reproducibility than the HC2 assay [28]. Owing to its easy application and good reproducibility, with large sample sizes, cobas^®^ 4800 has been implemented for cervical cancer screening in many countries [29,30,31]. The reverse line blot hybridization-based assay has demonstrated its ability to detect low risk (LR) and HR-HPV types in up to 40 samples, and approximately 40 HPV types simultaneously [32]. The REBA HPV-ID^®^ assay developed an automated machine for the hybridization step; however, DNA extraction and PCR amplification steps were performed manually. Therefore, HR-HPVs detection of REBA HPV-ID^®^ assay was evaluated. We compared the FDA-approved real-time-PCR-based cobas^®^ 4800 assay, which can detect 14 HR-HPV types including HPV16, 18 and additional 12 HR-HPV types, with the reverse line blot hybridization-based assay REBA HPV-ID^®^, which can detect both HR- and LR-HPV. We further compared the diagnosis rates of CIN2/3 lesions. 

## 2. Materials and Methods

### 2.1. Clinical Specimens

Cervical cell samples, collected from 303 women at the King Chulalongkorn Memorial Hospital (Department of Obstetrics and Gynecology, Faculty of Medicine, Chulalongkorn University, Bangkok, Thailand), were used for HPV-DNA detection and genotyping using the cobas^®^ 4800 HPV and REBA HPV-ID^®^ assays. The HPV-DNA results were compared to histology diagnoses. The study was approved by the Institutional Review Board (IRB) of Faculty of Medicine, Chulalongkorn University (COA No. 278/2019). 

### 2.2. Cobas^®^ 4800 HPV-DNA Detection and Genotyping

We used the fully-automated real-time PCR-based cobas^®^ 4800 HPV assay (Roche, Switzerland) for HPV detection and genotyping. The assay can detect HPV16, HPV18, and the pooled presence of twelve other HR-HPVs (HPV31, 33, 35, 39, 45, 51, 52, 56, 58, 59, 66 and 68). The reaction uses a β-globin-primer set as a genomic DNA control to ensure sample quantity and quality. In terms of sensitivity, the limits-of-detection for each HPV type per ml of the original sample were as follows: HPV16 were 85.0% for 150 copies per ml and 100% for 300 copies per ml; for HPV18, these were 93.2% for 300 copies per ml and 100% for 600 copies per ml [33]; for types 31, 33, 35, 39, 45, 58, 59 and 68, the limit-of-detection was 100% for 150–300 copies per ml; and for types 51, 52, 56 and 66, it was 100% for 1200–4800 copies per ml. No false positives were detected when the assay was performed using a panel of commonly found urogenital tract organisms, including bacteria such as *Lactobacillus* species, *Neisseria* species, *Staphylococcus* species, *Chlamydia trachomatis* and *Gardnerella vaginalis*, fungi such as *Candida albicans*, viruses such as HSV1, HSV2, CMV and EBV, and other LR-HPV types such as, HPV6 and HPV11. These indicated that the cobas^®^ 4800 assay specifically detects only HPV16,18 and other 12 HR-HPVs. The limit of detection and the analytical specificity written in the manuscript were determined by the manufacturer.

### 2.3. REBA HPV-ID^®^ HPV-DNA Detection and Genotyping

The distribution of HPV types was evaluated using the REBA HPV-ID^®^ assay (Molecules and Diagnostics, Wonju, Republic of Korea). REBA can identify 32 HPV types: 18 HR types (HPV16, 26, 18, 31, 33, 35, 39, 45, 51, 52, 53, 56, 58, 59, 66, 68, 69, 73), 1 probable HR type (HPV34), and 13 LR types (HPV 6, 11, 32, 40, 42, 43, 44, 54, 70, 72, 81, 84, 87). The membrane strip hybridization control line verifies the chromogenic reaction and the presence of the reagents; the negative control line checks for contamination; and the β-globin positive control line verifies the successful PCR amplification and quality of collected samples. 

We conducted HPV genotyping as per the manufacturer’s instructions. Briefly, extracted DNA was amplified in a 50 μL PCR reaction volume (25 μL 2× PCR premix, 16 μL DNase/RNase-free water, 4 μL primer mix, and 5 μL DNA). We ran the positive control (HPV33-positive) and negative control provided with the kit in parallel. The PCR conditions were as follows: initial denaturing at 94 °C for 5 min, followed by 15 cycles of 94 °C for 30 s, 55 °C for 30 s, followed by 45 cycles of 94 °C for 30 s, 52 °C for 30 s, and 72 °C for 10 min. The PCR condition was similar to the previous report with some modifications [34]. HPV genotyping was performed using the automated REBA HPV-ID^®^ system. The limits of detection were 100 to 1000 copies per PCR reaction (50 μL). There was no cross-reactivity between HPV types. The limit of detection and the analytical specificity were determined by the manufacturer.

### 2.4. Statistical Analysis

Statistical analysis was performed using SPSS for Windows 22.0 (SPSS Inc., Chicago, IL, USA). We evaluated HPV genotyping concordance between the cobas^®^ 4800 and REBA HPV-ID^®^ HPV assays using the kappa statistic, which ranges from 0 to 1: <0.20, poor; 0.21–0.40, weak; 0.41–0.60, moderate; 0.61–0.80, good; and 0.81–1.00, very good. Two-sided *p*-values were calculated via McNemar’s chi-square (*p* < 0.05).

## 3. Results

### 3.1. Overall Findings

HPV-DNA genotyping and histological investigation were performed on 303 cervical cell and colposcopy tissue samples, respectively. Histological examination revealed that 177 (58.4%) were positive for cervical intraepithelial neoplasia 1 (CIN1), 14 (4.6%) for CIN2, and 46 (15.2%) for CIN3, with 66 (20.8%) cases of other conditions, namely vaginal intraepithelial neoplasia, chronic cervicitis, benign squamous epithelium, and condyloma acuminata. The average age of the women was 40.76 ± 12.196 years (range, 18–75 years). 

### 3.2. Comparison of Cobas^®^ 4800 and REBA HPV-ID^®^ Assays

Of the 303 cases, cobas^®^ 4800 and REBA HPV-ID^®^ identified 72.27% (219) and 71.62% (217) HR-HPV positive cases, respectively. 

Among all of the HR-HPV cases, cobas^®^ 4800 and REBA HPV-ID^®^, respectively, identified 71.19% (126) and 70.06% (124) of the 177 CIN1 cases; 100% (14) and 92.86% (13) of the 14 CIN2 cases; 78.26% (36) and 86.96% (40) of the 46 CIN3 cases; and 65.15% (43) and 60.60% (40) of the 66 non-CIN (other abnormality) cases (Table 1). The REBA HPV-ID^®^ assay identified the following HR-HPV types: HPV16, 18, 31, 33, 39, 51, 52, 53, 56, 58, 59, 66,68 and 73. 

The cobas^®^ 4800 and REBA HPV-ID^®^ assay, respectively, identified HPV16 infection in 35.64% (108) and 30.69% (93) of the 303 cases. Of the HPV16-positive cases, the detection rates were lowest for the CIN1 cases (28.81% and 25.42% for cobas^®^ 4800 and REBA HPV-ID^®^, respectively), highest for CIN2 (71.43% and 64.29, respectively), and intermediate for CIN3 (50% and 45.65%, respectively) (Table 1).

Of the LR-HPV types, the majority were found either in CIN1 (11.29%, 20/177), or other abnormalities (21.21%, 14/66) but less frequently found in CIN2/3 (6.67%, 4/60) (Table 1). The LR-HPV types identified by REBA HPV-ID^®^ were HPV6, 11, 32, 42, 43, 54, 70, 72, 81, 84 and 87.

The assays did not differ significantly in the HR-HPV positive case detection rate (*p* = 0.112; concordance rate, 87.45%; kappa = 0.689). For all HPV-positive cases, the assays differed significantly (*p* = 0.005), because cobas^®^ 4800 does not detect LR-HPV types (Table 2). Of the 45 cobas-negative/REBA-positive discrepant samples, 27 (60%) were LR-HPV, 14 (31.1%) were mixed LR- and HR-HPV types and only 4 (8.9%) were single HPV16. Probable HR-HPV types such as HPV53 and 73 detected by the REBA HPV-ID^®^ assay were not included in the other twelve HR-HPVs of the cobas^®^ 4800 assay. Therefore, the cobas^®^ 4800 assay detected only HR-HPV types, with no false positives among the LR-HPV-positive samples. Based on histological grading, the HR-HPV concordance of the assays was 88.70% for CIN1 and 88.33% for CIN2/3 (Table 3).

## 4. Discussion

For more than 50 years, cervical cancer screening has been done using cytology or Pap smear [35]. Women with abnormal cytology will be referred for colposcopy and biopsy will be taken for a histology examination. However, 5%–35.6% of ASCUS and LSIL cytology revealed histology diagnosed as CIN2+ [14,36,37,38]. To improve cervical cancer screening, HR-HPVs or oncogenic HPVs testing has been used as in combination with cytology, to specifically screen women who need further colposcopy examination [39]. In 2020, According to the American Cancer Society (ACS) guideline, the US-FDA approved HR-HPVs testing alone can be used for cervical cancer screening, it is recommended that women begin cervical cancer screening at the age of 25 [40]. The cobas^®^ 4800 assay has been approved by the US-FDA for use as a stand-alone assay for the screening of cervical cancer [41]. The cobas^®^ 4800 assay has been clinically evaluated and has significant sensitivity for the detection of CIN2+ lesions [20,21,42]. Moreover, all steps of the cobas^®^ 4800 assay could be completed on a fully automated machine. Among various HPV assays, other non-US-FDA approved assays based on real-time PCR detect only HR-HPVs [41]. To detect LR-HPVs, the PCR-reverse line blot hybridization-based assay (REBA HPV-ID^®^) was utilized in the present study to detect both HR-and LR-HPVs; additionally, the hybridization step could be performed using an automated machine. In terms of performance capacity, the automated cobas^®^ 4800 assay can complete 96 tests, including 94 clinical samples and 2 controls, in roughly five hours. While, 48 samples can be run through the REBA HPV-ID^®^ assay in about 6 hours, the DNA extraction, PCR amplification, and automated hybridization were done separately. Prior to PCR amplification, samples could be conveniently prepared by adding lysis buffer and boiling [34,43]. To evaluate the HR-HPV detection of REBA HPV-ID^®^ assay, we compared an US-FDA-approved real-time PCR-based method that detects only HR-HPVs (cobas^®^ 4800), and a reverse line blot hybridization-based assay (REBA HPV-ID^®^) that detects both LR- and HR-HPVs. Because of this difference in their detection ability, the assays differed significantly in detecting HPV-positive cases. The differences we observed verify that there was no cross-reactivity (false-positive LR-HPV detection) by the cobas^®^ 4800 assay, which is consistent with a previous report [28].

The assays showed a strong concordance for HR-HPV detection, although for CIN2/3-positive cases, the small sample size (n = 60) resulted in a low kappa value. Single or mixed HPV16 infections were detected better by cobas^®^ 4800 than REBA HPV-ID^®^, consistent with prior findings [42,44,45]. The cobas^®^ 4800 assay showed high sensitivity for CIN2+ detection, relative to prior cytology results [20,42]. To the best of our knowledge, all prior studies of the REBA HPV-ID^®^ assay have used samples with cytologically confirmed lesions [43,46].

For the HPV16 type, mostly detected in CIN2/3 cases, with lower detection rates for CIN1 and other abnormality cases, consistent with earlier investigations [47,48,49,50]. Here, the REBA HPV-ID^®^ assay identified the following CIN2/3-positive HR-HPV types: HPV16, 18, 31, 33, 51, 52, 56, 58, 66, 59, and 68, which is similar with the previous study [51]. Among the LR-HPV types, the most commonly detected cases were those that were diagnosed as CIN1 and other cervical cell abnormalities. Our finding revealed only 6.67% of CIN2/3 cases were found to have LR-HPV infections, which is consistent with earlier findings [52,53]. If LR-HPV types are suspected in samples with high-grade cervical lesions that are HR-HPV type negative, they can be detected using the REBA HPV-ID^®^ assay. LR-HPV types cause benign proliferative lesions such as genital warts, as well as recurrent respiratory papillomatosis (RRP), and is not frequently found in cervical cancer [54].

In conclusion, our findings show that the clinically evaluated US-FDA-approved HR-HPV detection assay could better detect an HPV16 infection and is appropriate for screening women with persistent HR-HPV infections, and for predicting the increased risk of cervical cancer progression. The REBA HPV-ID^®^ assay can be used in the HPV types distribution for epidemiological studies.

## Figures and Tables

**Table 1 viruses-14-02713-t001:** HPV DNA detection by the cobas^®^ 4800 and REBA-HIV-ID^®^ assays, stratified histologically, in 303 samples.

HPV Results	Methods	Histology				Total
		CIN1	CIN2	CIN3	Non-CIN	
Total number	cobas/REBA	177	14	46	66	303
HPV negative	cobas	51(28.8%)	0(0.0%)	10(21.7%)	23(34.8%)	84(27.7%)
	REBA	33(18.6%)	0(0.0%)	3(6.5%)	12(18.2%)	48(15.8%)
HPV positive (including HR-and LR-HPV)	cobas	126(71.2%)	14(100.0%)	36(78.3%)	43(65.2%)	219(72.3%)
	REBA	144(81.4%)	14(100.0%)	43(93.5%)	54(81.8%)	255(84.2%)
HR-HPVs (including HPV16 and 18)	cobas	126(71.2%)	14(100.0%)	36(78.3%)	43(65.2%)	219(72.3%)
	REBA	124(70.1%)	13(92.9%)	40(87.0%)	40(60.6%)	217(71.6%)
Single HPV16	cobas	19(10.7%)	6(42.9%)	12(26.1%)	13(19.7%)	50(16.5%)
	REBA	21(11.9%)	4(28.6%)	12(26.1%)	11(16.7%)	48(15.8%)
Mixed HPV16 & other HR-HPVs	cobas	32(18.1%)	4(28.6%)	11(23.9%)	11(16.7%)	58(19.1%)
	REBA	24(13.6%)	5(35.7%)	9(19.6%)	7(10.6%)	45(14.9%)
Single HPV18	cobas	5(2.8%)	0(0.0%)	2(4.3%)	1(1.5%)	8(2.6%)
	REBA	2(1.1%)	0(0.0%)	4(8.7%)	1(1.5%)	7(2.3%)
Mixed HPV18 & other HR-HPVs	cobas	0(0.0%)	0(0.0%)	0(0.0%)	1(1.5%)	1(0.3%)
	REBA	4(2.3%)	0(0.0%)	0(0.0%)	1(1.5%)	5(1.7%)
Other HR-HPV (HPV16/18 not included)	cobas	70(39.5%)	4(28.6%)	11(23.9%)	17(25.8%)	102(33.7%)
	REBA	73(41.2%)	4(28.6%)	15(32.6%)	20(30.3%)	112(36.9%)
LR-HPVs (including untyped HPVs)	cobas	NA	NA	NA	NA	0
	REBA	20(11.3%)	1(7.1%)	3(6.5%)	14(21.2%)	38(12.5%)
Total HPV16 (single + mixed infections)	cobas	51(28.8%)	10(71.4%)	23(50.0%)	24(36.4%)	108(35.6%)
	REBA	45(25.4%)	9(64.3%)	21(45.7%)	18(27.3%)	93(30.7%)
Total HPV18 (single + mixed infections)	cobas	5(2.8%)	0(0.0%)	2(4.4%)	2(3.0%)	9(3.0%)
	REBA	6(3.4%)	0(0.0%)	4(8.7%)	2(3.0%)	12(4.0%)

**Table 2 viruses-14-02713-t002:** Comparison of HPV detection between the cobas^®^ 4800 and REBA-HIV-ID^®^ assays.

Methods		Cobas		Total	Percentage Agreement	Kappa Value
		HPV Positive	HPV Negative			
REBA	HPV positive ^1^	210	45	255	82.17	0.488 ^3^
	HPV negative	9	39	48		
	Total	219	84	303		
REBA	HR-HPV positive	199	18	217	87.45	0.689 ^3^
	HPV negative ^2^	20	66	86		
	Total	219	84	303		

^1^ Including HR- and LR-HPV; ^2^ Including LR-HPV positive; ^3^ The kappa statistic ranges from 0 to 1 (<0.20, poor; 0.21–0.40, weak; 0.41–0.60, moderate; 0.61–0.80, good; and 0.81–1.00, very good).

**Table 3 viruses-14-02713-t003:** HR-HPV detection rate comparison between the cobas^®^ 4800 and REBA-HIV-ID^®^ assays, stratified histologically.

Histology	Methods		Cobas		Total	Percentage Agreement	Kappa Value
			HR-HPV Positive	HPV Negative			
Other	REBA	HR-HPV positive	36	4	40	83.33	0.49 ^2^
abnormality		HPV negative ^1^	7	19	26		
		Total	43	23	66		
CIN1	REBA	HR-HPV positive	115	9	124	88.70	0.76 ^2^
		HPV negative ^1^	11	42	53		
		Total	126	51	177		
CIN2/3	REBA	HR-HPV positive	48	5	53	88.33	0.51 ^2^
		HPV negative ^1^	2	5	7		
		Total	50	10	60		

^1^ Including LR-HPV positive; ^2^ The kappa statistic ranges from 0 to 1 (<0.20, poor; 0.21–0.40, weak; 0.41–0.60, moderate; 0.61–0.80, good; and 0.81–1.00, very good).

## Data Availability

The data presented in this study are available upon reasonable request from the corresponding author.

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
