# Peer review of "High-Risk Human Papillomavirus Detection via Cobas® 4800 and REBA HPV-ID® Assays"

_viruses, 2022, doi:10.3390/v14122713_

Round 1

Reviewer 1 Report

Manuscript details:
Journal: Viruses
Manuscript ID:
viruses-2028574
Type of manuscript: Article
Title: High-risk human papillomavirus detection via cobas® 4800 and REBA HPV-ID® assays

In the manuscript entitled “High-risk human papillomavirus detection via cobas® 4800 and REBA HPV-ID® assays”.  Major comments to the author should be concrete and helpful for revision and may include overall strengths and weaknesses of the scientific merit and research approach. A few comments that should be addressed before this manuscript could be considered for publication are provided below. In addition, I would like to minor revisions before publication. Below are the suggestions to improve the manuscript.

I suggest don't repeat title words in keywords.

L37: Please use “HPV-16 or HPV16” in anyone style in the whole manuscript.

L70-74: Rewrite and try to clarify the meaning

In the cobas assay author used “copies per ml” and the REBA assay used “copies per reaction” for limit-of-detection, why is the difference?

How did the author determine the limit-of-detection?

L 93-95: No false positives were detected when the assay was performed using a panel of commonly found urogenital-tract organisms, including bacteria, fungi, viruses, and other LR-HPV types…. please explain  

L109-112: Use the PCR condition citation

L180: Remove double space “ CIN2+  detection”

L189; Remove one full stop “ . . If LR-HPV”:

What are the advantages of these two methods? In other words, how these two strategies are superior to other methodologies? The authors should discuss this.

 The discussion part is too short, and more efforts should be made to interpret the data.

Please try to clear the presentation in Tables.

Author Response

Subject: Resubmission of revised manuscript (viruses-2028574)

Dear Reviewer

We would like to thank the reviewers for the valuable comments and suggestions, which help to improve the quality of the manuscript. The adjustments have been carefully done in the revised manuscript. The explanations or clarifications of each point raised are detailed as attached file.

We very much hope the revised manuscript is accepted for publication in Viruses Journal.

Best regards,

Arkom Chaiwongkot, Ph.D

Reviewer 2 Report

This paper compares the HPV detection rates of Cobas 4800 and REVA HPV-ID assays. There is no difference in the detection rate of high-risk HPV between Cobas 4800 and REVA HPV-ID assays are particularly useful for detecting low-risk HPV. The content is adequate.

Minor modifications should be made as follows
Line 85, RT-PCR
If Real-time PCR is written as RT-PCR, it should be noted as Real-time PCR (RT-PCR) first so that the abbreviation is unambiguous.

Author Response

(The authors gave the same response as above.)
